# Antibiotic Resistance in the Finfish Aquaculture Industry: A Review

**DOI:** 10.3390/antibiotics11111574

**Published:** 2022-11-08

**Authors:** Gianluigi Ferri, Carlotta Lauteri, Alberto Vergara

**Affiliations:** Faculty of Veterinary Medicine, Post-Graduate Specialization School in Food Inspection “G. Tiecco”, University of Teramo, Strada Provinciale 18, 64100 Teramo, Italy

**Keywords:** aquaculture, finfish, antibiotic molecule, molecular biology, one-health

## Abstract

Significant challenges to worldwide sustainable food production continue to arise from environmental change and consistent population growth. In order to meet increasing demand, fish production industries are encouraged to maintain high growth densities and to rely on antibiotic intervention throughout all stages of development. The inappropriate administering of antibiotics over time introduces selective pressure, allowing the survival of resistant bacterial strains through adaptive pathways involving transferable nucleotide sequences (i.e., plasmids). This is one of the essential mechanisms of antibiotic resistance development in food production systems. This review article focuses on the main international regulations and governing the administering of antibiotics in finfish husbandry and summarizes recent data regarding the distribution of bacterial resistance in the finfish aquaculture food production chain. The second part of this review examines promising alternative approaches to finfish production, sustainable farming techniques, and vaccination that circumvents excessive antibiotic use, including new animal welfare measures. Then, we reflect on recent adaptations to increasingly interdisciplinary perspectives in the field and their greater alignment with the One Health initiative.

## 1. Introduction

Recent reports indicate that finfish and seafood consumption can be considered sustainable feeding sources. In a recent FAO Report (2020), it was observed that 156 million tons were for human consumption; this means roughly 20.5 kg/consumer per year. The remaining amount (23 million tons) was used for fish oil and fishmeal production [1]. In 2015, the world marine catch decreased by almost 2 million tons from 81.2 million tons. This decrease was justified by significant fish catching, which has led to reduced animal densities, causing environmental alterations due to the high impacts of anthropic activities [2].

Generally, fisheries have strategic importance for food production, human and animal nutrition, and the employment of millions of people (39 million people in the primary sector of capture fisheries and 20.5 million people in the aquaculture one) [3].

With the reduction in marine finfish populations and the increase in human consumers, aquaculture farmers worldwide have expanded productive systems from small fisheries to larger, more intensive ones; many countries (i.e., China, Thailand, India, etc.) have promoted the building of inland and mariculture husbandries [1,4]. In 2018, most reports indicated that global fish production had increased from 167 million tons in 2016 to 179 million tons, with 82 million tons being derived from aquaculture farms [1]. The intensively farmed finfish species present high mortality (50%), starting from the larval stage and continuing in sea cages. Losses are generally caused by bacterial or viral diseases; in major cases, bacterial pathologies are directly linked to the high density located in feces and sediments or to improper vaccination programs [4].

The high animal density, directly related to the welfare concept, is a crucial aspect because it provides epidemiological and environmental conditions that lead to possible infectious disease outbreaks that cause productive and economic losses. In most cases, antibiotics usually represent the first choice for treatments [5], which can be administered through different routes: feed, water immersion, or injection [6]. From a zootechnic perspective, it is also relevant to considerer two possible negative aspects: proper usage and the administering of unapproved (antibiotics in which usage is restricted only to the human medicine) or illegal molecules [i.e., chloramphenicol (banned in the EU member states)].

Comparing aquaculture to the terrestrial farms, based on pharmacological consumption, researchers have highlighted significant differences. Indeed, the World Health Organization classified aquaculture as an activity with a low environmental impact for antibiotic usage [7,8]. Directly linked to the above-explained concepts, veterinarians play key roles in pharmacological management. This consideration is justified because this professional figure firstly prescribes antibiotic therapies, avoiding the unnecessary administering (as metaphylactic one) of certain classes—in which usage is restricted to the human medicine: the so-called Critical Importance Antimicrobials WHO (CIA)—for food-producing animals; secondly, they must reduce administering to only restricted specific cases. The aim is to decrease a relevant selective pressure, which promotes resistant and pan-resistant bacterial strains survival [1]. These strains are named as antibiotic resistance bacteria (ARBs) that can be considered as “*drivers*” of antibiotic resistance genes (ARGs) with important repercussions on the environmental, animal, and human health [9]. It has been widely demonstrated that ARGs can be transferred to human intestinal microbiota and, consequently, to the ingestion of foods (numerous matrices: meat, dairy products, fish, etc.), which can drive commensal or pathogenic (*Salmonella* spp., *Vibrio* spp.) ARBs with extra chromosomal resistance forms. Finally, it is important to mention possible drugs residues due to the improper observation of legal limits [7]. Therefore, the European Regulations No. 470/2009 and No. 37/2010 established the residual limits of pharmacologically active substances in animal origin foodstuffs.

Furthermore, the aquatic environment (i.e., oceans, lakes, rivers) is also a possible reservoir of ARGs [1]. Generally, fin fish’s intestinal microbiota is characterized by bacterial populations that are like those ones detected in the aquatic environment. Therefore, microbiological water quality (influenced by wastewater management, fish industries, and other anthropic activities, etc.) can represent a critical environmental resistance factor that allows ARGs diffusion and preservation [1].

## 2. Antibiotic Usage: Regulations in Aquaculture Farms

A horizontal concept, that involves, at the same time, different cultured animal species, is represented by the therapeutic administering of antibiotic molecules related to the high subjects’ densities (per m^2^ or m^3^ of surface or water). Due to its crucial aspect, represented by the antimicrobial resistance (AMR) phenomenon, different nations have organized their respective legislations to prevent and decrease its diffusion. The WHO data report an alarmistic scenario: within 10 years, the antibiotic therapeutic efficacy, both from humans and animals, will be strongly reduced [7].

In this section, authors want to describe the main legislative measures adopted by different nations. During last 20 years, European and American (USA) public health institutions have produced lists of authorized molecules [10].

In 2000, the European Union, firstly, in the “*White Paper on Food Safety*”, identified the strict correlations between food and environmental safety concepts. From this document, the European Commission has evolved and, on this issue, has based the new regulations: EU Reg. No. 178/2002; EU Reg. No. 852/2004; EU Reg. No. 853/2004; EU Reg. No. 625/2017 [11].

The above-mentioned regulations supplement obligatory requirements for producing countries to follow the Council Directive 96/23/EC for aquaculture products to export to the EU States [12]. These legislative acts are supported by strict monitoring activities regarding usage and trade of veterinary antibiotics performed by law-designed control figures, i.e., the European Surveillance of Veterinary Antimicrobial Consumption (ESVAC) [13]. Indeed, the European Medicines Agency (EMA) banned the administering of certain molecules (cefuroxime, chloramphenicol, polymyxin B Sulphate, and Nystatin) to guarantee final consumer health [14]. More recently European agencies, including the European Commission, established maximum residue limits (MRL) of pharmacologically active substances in foodstuffs of animal origin (including finfish): EU Reg. No. 470/2009, and No. 37/2010 (see Table 1).

Moreover, the Food and Drug Administration (FDA) and the United States Department of Agriculture (USDA) worked on fish drugs, reporting a list of antibiotics that can be used in aquaculture [5]. As described above by Table 1 and Table 2, comparing developed countries legislations, the European limit values of active substances are expressed as µg/kg and mg/kg in the American one. Furthermore, in European legislation (i.e., EU Reg. No. 37/2010), more detailed and species-specific (including finfish and shellfish) limit values concerning a wide range of xenobiotic molecules than the North American ones are reported.

In the developing countries, there is a wide differentiation, which depends on their respective governmental agencies. This last sentence should not be considered redundant, because a clear legislation on antibiotic usage in veterinary medicine, more specifically on the aquaculture sector, is not yet well structured.

For the above-mentioned reasons, and in order to export finfish products, developing countries’ legislators adopted similar parameters to those reported in the European Union and USA laws, i.e., the Brazil [15], Vietnam [16,17,18], Chile [19], China [20,21,22,23], India (MPEDA) [23], the Philippines [1], and Thailand [24].

In China, Thailand, Vietnam, Brazil, Chile, Bangladesh, Norway, the Philippines, and India, governmental authorities have listed the authorized antibiotic compounds and banned other molecules for usage in the aquaculture sector [16,17,20,24,25,26]. In Asia, there are differences between geographic regions, depending on the antimicrobial usage (any farmers that still administer chloramphenicol that is banned in the aquaculture zootechnic) and local food safety regulatory agencies. However, this last issue has been justified by low pharma-surveillance programs, poor “*Food Safety Legislation*”, and inadequate monitoring systems’ control of drug usage [27]. These legal and surveillance gaps have contributed to the ARBs and ARGs in the aquatic environment [28,29,30].

To contrast the AMR phenomenon, there are multiple examples of collaboration between different legislative institutions that have improved the antimicrobial management. For instance, the FDA continues to detect nitrofurans and chloramphenicol in collaboration with Malaysian aquaculture producers, and, consequently, this country has banned them [31]. China realized innovative and specific “*Applicative Guidelines*”, which specify sulfonamides, tetracycline, and enrofloxacin usages, which were adopted in other Asian countries, i.e., Vietnam [23].

In conclusion, it is possible to affirm that the pleomorphic AMR phenomenon cannot be reduced by geographical limits, and the respective legislations difficulty could be organized with the same restrictions [11]. However, due to the environmental implications, which pose at-risk human and animals health, it is mandatory to align lists of antibiotic molecules that can be used for finfish disease treatments. Therefore, sanitary authorities in the international community should make an effort to contribute to a global reduction in their tons of consumption of finfish production, starting from the sharing of data and enforcing innovative pharmaco-surveillance systems. It implies that the realization of integrated tracing processes, which have origins from the pharma industries, arrive at the administering step in aquaculture farms and, consequently, involve the aquatic environment.

## 3. Aquatic Environment and Antibiotic Resistance Circulation

Infectious disease caused by ARBs are estimated to cause 10 million deaths worldwide by 2050 [32]. This issue is a horizontal problem involving humans, animals, plants, foods, and environments; due to these reasons, the One Health approach is essential to overcome this developing threat [33].

Previous studies have identified the aquatic environments (which include oceans, lakes, and rivers) as potential transmission routes and ARGs and ARBs reservoirs [10,24,33]. Water and, in particular, wastewater management are crucial steps in the so called “*water-cycle*” as vectors of antibiotic resistance forms. Indeed, it has been repeatedly reported that ARGs and ARBs detection in water samples collected from treatment plants and in numerous cases the public sanitary authorities (in accordance with specific national legislation cut-offs) have found high residual titers of antibiotic molecules (i.e., quinolones, tetracyclines, carbapenems, aminoglycosides, etc.) [34].

For this purpose, the usage of specific filters for wastewaters management could be useful for their contributions to the reduction in the environmental diffusion of microorganisms’ loads (bacteria, virus, etc.) [35]. Many research studies have analyzed the anthropic impact generated by hospitals, farms, domestic environments, and food industries by evaluating 79 wastewater samples in different geographic areas, identifying a limited AMR cluster encoding resistance against macrolides, quinolones, aminoglycosides (more than 30% of screened samples) in developed countries (Europe, North America, and Oceania). However, in developing continents (Asia, Africa, and South America) ARGs were mainly reported to encode resistance against sulfonamides and phenicols (especially chloramphenicol (40% of samples)) [34].

Animal origin manures, largely used in the agricultural sector, can hide notable risks for humans. It has been widely demonstrated that their fertilization usage is a reasonable source of further ARGs environmental diffusion. These substances are responsible for aquifers contaminations becoming an environmental concern and providing tangible evidence, where terrestrial and aquatic productive realties are strictly influenced [36]. Hatosy and Martiny [37] evidenced that the 28% of detected ARGs in marine water samples were transferred by freshwater and wastewaters flows. These findings also provide scientific evidence that the coastal runoff from terrestrial sources is one of the ARGs mechanisms of diffusion.

The selective pressures, caused by different anthropic activities, have further repercussions on mariculture farms, as demonstrated by Miranda et al. [38]. They discovered, in different Chilean salmonid farms, high circulation of tetracycline and quinolones ARGs. Authors justified these findings by the large antibiotic administering reported by public health institutions: an amount 363.4 antimicrobial tons were used by farmers [38]. To improve knowledge about this phenomenon, many researchers have studied other possible ARGs and ARBs reservoirs. Muziasari et al. [39] focused on the role of sediment and fish feces collected from mariculture and inland farms in the Baltic Sea. They observed ARGs presence in intestinal contents from rainbow trout (*Oncorhynchus mykiss*) specimens collected in different aquaculture systems. They discovered different resistance determinant amounts using the real time PCR assay: tetracycline and, in particular, *tet genes* (*tet*M: 6.25 10^−2^ copies); aminoglycosides target genes *erm* (*erm*B 3.13 10^−1^ copies); and sulfonamides, such as *sul* (*sul*3 3.13 10^−1^ copies). Furthermore, similar patterns were amplified from sediment samples. The phylogenetic analysis allowed us to demonstrate the same genomic source. These findings highlighted and enforced a fundamental concept concerning the antibiotic resistance phenomenon in which animal and environmental microbiomes are strictly connected to each other through horizontal gene transmissions.

From these considerations, the so-called *One Health* approach results are mandatory. The aquatic environment, characterized by various bacterial strains, can be considered as possible drivers of resistant forms. The aquatic creatures, including finfish ones, can be considered as entropic systems, where the intestinal microbial populations meet environmental ones involved in a fascinating “*antibiotic resistant genes trade”*. In this articulated “*ARGs life-cycle*”, final human consumers microbiomes could also be involved by the fish food commensal strains, harboring resistance genes; they can transmit them to the human intestinal bacterial species. These conditions are realistically responsible for the emergence of multidrug-resistant (MDR) or pan-resistant pathogenic or commensal microorganisms’ spreading [40].

In Asia, public sanitary authorities reported that fraudulent antimicrobial administering by farmers (in the order of tons) have conducted the selection of MDR microorganisms (*Escherichia coli*) isolated from aquaculture finfish. Phylogenetic analysis also demonstrated that other species belonging to the family *Enterobacteriaceae*, isolated from river water samples, presented the same phenotypic and genotypic resistance pattern and same codifying sequences [1].

Although antimicrobial molecules are administered in inland farms (closed ecosystems), the water’s turnover, performed by filter systems, represents a further potential source for environmental antibiotic diffusion, having critical repercussions on final consumers’ health [41]. Therefore, water’s microbiological quality is a crucial element that is influenced by different parameters and factors, i.e., temperature, salt content, space distance between coasts and catching areas (especially influenced by anthropic activities), natural presence of bacteria in the water environment, nutrition for fish, farming systems, catching methods, and technological aspects [1]. These parameters influence animal welfare (in accordance with the “*Animal Health Law*” EU Reg. No. 429/2016) and immunity, since stressed animals are more susceptible to infection and, in turn, require more application of antibiotic therapy. These molecules are especially used at larval stages, when farmers should register high mortality rates. To avoid this problem, they (farmers) often improperly administer antimicrobials, contributing to the ARBs and ARGs enforcement and diffusion.

From an ecological point of view, Reverter et al. [41] studied the effects and possible correlations between water temperature on aquatic animal mortality related to MDR microorganisms. They evaluated these aspects on bred aquatic animals (finfish), artificially infected with specific fish pathogens (i.e., *Vibrio* spp.). Researchers found a statistical significant correlation between high water temperature (simulating “*global warming*” phenomenon) and infected treated finfish. Warm water resulted responsible for high AMR diffusion. At the same time, they also discovered calculating multiple antibiotic resistance (MAR) indices through the phylogenetic analysis and metagenomic evaluations, a further statistical correlation to the bacteria isolated in human pathogenic specimens reservoirs. These findings were justified by the high human activities’ impact (agrochemical substances, toxic metals, abattoirs, and wastewaters managements) [42,43]. Furthermore, based on chemical characteristics, any molecules (including antibiotics) released in the marine environments are low bio-degradable and, for this reason, have been named as “*pseudo-persistent*” [44]. In polluted coastal areas, any researchers discovered a high direct correlation between heavy metal residue titers (as lead, cadmium, and mercury) and the horizontal ARGs transmission (more specifically observed for tetracycline molecules: *tet* genes) [45].

In conclusion, human-impacted aquatic areas (i.e., oceans, lakes, rivers, etc.) are responsible for the maintenance and diffusion of MDR bacteria, which have been defined as strains resistant to at least three antibiotic classes. For these reasons, they represent a public health issue [1]. In this way, the environment involves the role of one of the main reservoirs regarding MRB, and it represents “*le fil rouge*” between antibiotic residues (due to agricultural runoffs, sewage discharges, and leaching from nearby farms) and public health [46].

## 4. Global Antibiotic Administering in the Aquaculture Sector

The global antibiotic consumption is considered a dynamic value due to its annual variability. In 2017, scientists estimated that a total amount of 10,259 antibiotic tons were administered to food-producing animals [1]. The continent of Asia represents the largest producer and consumer (93.8%) of such molecules, larger than Africa (2.3%), and Europe (1.8%). Four countries present the highest consumption levels: China (57.9%), India (11.3%), Indonesia (8.6%), and Vietnam (5%) [11], as illustrated in Table 3.

EFSA and FDA have proposed a common aim that indicates an antibiotic usage reduction of 30% by 2030 [1]. Conversely to this purpose, there is an opposite trend reported in the BRICS countries, Brazil, Russia, India, China, and South Africa, where scientists estimate an increasing antibiotic consumption for terrestrial food-producing animals. The estimated total amounts could exceed human use [47]. This prospect for the next future has been also confirmed by Schar et al. [48] (Brazil (94%), Saudi Arabia (77%), Australia (61%), Russia (59%), and Indonesia (55%); these percentages are strictly correlated to the respective national usages) (See Table 3).

**Table 3 antibiotics-11-01574-t003:** Antibiotics and their distribution in finfish aquaculture from different geographical regions.

Continents	Countries	Antibiotic Classes
Asia-Pacific: 9623 tons	China: 5.572 tons [49]	Tetracyclines: 3065 tons
		Quinolones: 1393 tons
		Beta-lactams: 836 tons
		Sulfonamides (co-administered with phenicols): 278 tons
	India: 1.087 tons [48]	Tetracyclines: 706 tons
		Beta-lactams: 195 tons
		Quinolones: 186 tons
	Indonesia: 827 tons [48]	Tetracyclines: 645 tons
		Beta-lactams: 182 tons
	Vietnam: 481 tons [48]	Tetracyclines: 370 tons
		Quinolones: 62 tons
		Beta-lactams: 49 tons
Africa: 236 tons [48]	Egypt: 110 tons	Tetracyclines: 86 tons
		Beta-lactams: 13 tons
		Quinolones: 11 tons
	South Africa: 126 tons	Tetracyclines: 107 tons
		Sulfonamides: 19 tons
Europe: 185 tons	Turkey: 75 tons [50]	Tetracyclines: 39 tons
		Beta-lactams: 16 tons
		Quinolones: 8 tons
		Sulfonamides: 7 tons
		Phenicols (Chloramphenicol): 5 tons
	Norway: 45 tons [51]	Tetracyclines: 30 tons
		Sulfonamides: 10 tons
		Quinolones: 5 tons
	Scotland: 32 tons [51]	Tetracyclines: 28 tons
		Beta-lactams: 4 tons
	Italy: 13 tons [51]	Tetracyclines: 7 tons
		Beta-lactams: 4 tons
		Sulfonamides: 2 tons

The International Pharma Agencies report that global animal antimicrobial administering and consumption, and more specifically for the aquaculture sector, involve the following finfish species: 8.3% catfish (*Ameiurus melas*), 3.4% tilapia (*Tilapia* spp.), 2.7% shrimp (*Penaeus* spp.), 0.8% trout (*Oncorhynchus mykiss*), and 0.7% salmon (*Salmo salar*) [48].

Concerning the global antibiotic consumption volumes, innovative digital tracing systems will involve fundamental roles in the pharma-surveillance programs. Indeed, it is difficult to provide global monitoring with standardized data due to the multiple variables that can influence the results (i.e., legal antimicrobial administering, anthropic environmental impact, pollution, microbiological water quality, etc.) [52]. All these mentioned affirmations are examples of future challenges that cannot be postponed anymore.

Generally, the most frequently prescribed and detected antimicrobials are the following antibiotic classes: quinolones, tetracyclines, sulfonamides, and amphenicols [11,27].

Among the quinolone class, enrofloxacin, nalidixic acid, and ofloxacin are widely administered due to their chemical characteristics that permit to these molecules to be stable in water and sediment [52], resulting in them being easy to manage in the aquaculture [53]. There are also two different scenarios: in the developed countries, Lulijwa and coworkers found that 55% of global major aquaculture-producing countries have used enrofloxacin and less frequently ciprofloxacin and norfloxacin [11]. Indeed, since 2015 in China, the administering of norfloxacin has been banned for aquaculture [23].

Tetracyclines represent another important antibiotic class widely used in aquaculture farms due to their low costs in association with their high efficacy as a broad spectrum for treatment and prevention of infectious disease [10,54]. The WHO reported that doxycycline, oxytetracycline, and chlortetracycline have been administered for a long time in finfish farms. Due to the increasing antimicrobial resistance patterns, the WHO has suggested a further restriction of these molecules for veterinary usage [55,56]. More specifically, in Asia, oxytetracycline is the most commonly allowed-by-law and administered molecule in the aquaculture farms, and several studies have detected residues in water samples in numerous countries. Oxytetracycline residues have also been detected in the European water and sediment specimens, although, since 2006, it has been banned by all EU member states [57,58]. These considerations are justified by chemical characteristics, which confer a higher environmental resistance to oxytetracycline than the other molecules (belonging to the same antibiotic class) [59,60]. Regarding the tetracyclines, scientists estimated the following persistence periods: 21–25 min in aquaculture water, 2 days in freshwater, 12 days in seawater, 150 days in marine sediment (depending on chemical and environmental parameters as pH, temperature, salinity, and light) [61,62].

The sulfonamides class is largely administered in the finfish farms, and, in particular, veterinarians have prescribed sulfamethoxazole or the combined form sulfamethoxazole/trimethoprim. Generally, these molecules represent the third most prevalent antibiotic used in aquaculture after tetracyclines and quinolones [27]. They are largely used due to their low costs, high water solubility, and due to the high floating characteristics, allowing easy transport and distribution in the aquatic environments and adsorbed by finfish through the gills [63].

## 5. ARBs Isolation and ARGs Detection from Aquaculture Finfish Samples

The AMR phenomenon has been generally defined as the failure of growth’s inhibition or the killing capacity of an antimicrobial molecule beyond the normal susceptible bacteria [32,64].

The finfish aquaculture zootechnic sector has been characterized by a wide range of farming techniques as the embankment ponds or the watershed ones (as observed in the catfish (*Clarias* spp., *Ictalurus* spp., and *Pangasius* spp.) culture) [65], mariculture systems (i.e., for *Salmo* spp., *Sparus* spp., [66], intensive or semi-intensive inland pond systems for *Tilapia* spp. [65]), and other animals finfish species, etc.

In the above-mentioned systems the high animal densities have induced the necessity of antibiotic administering for therapeutic purposes. This last-explained concept was associated with possible inappropriate usages and has selected resistant pathogens or commensal bacterial strains. More in detail, tetracyclines, beta-lactams, quinolones, and sulfonamides antibiotic classes have been largely prescribed by veterinarians. Therefore, biomolecular diagnostic procedures have coupled the next generation sequencing to the bacterial whole genome analysis. This last cited method has permitted us to discover new oligonucleotide resistance determinants [63]. Oligonucleotide sequences are the main actors involved in the ARGs circulation and are, consequently, responsible for the presence of vector bacteria (not usually resulting in pathogens for humans) while constituting crucial environmental reservoirs for the human and animal host microbiota [64]. For these reasons, animal origin foodstuffs have acquired more attention from scientists. The reasons were firstly related to possible residual concentration, but more specifically the main concern is represented by the possibility of horizontal resistance genes transmission between alimentary commensal and opportunistic strains with the human microbiota. Indeed, microorganisms have elaborated numerous mechanisms to disseminate the ability to survive by mobile genetic elements, such as integrons, plasmids, insertion sequences, transposons, and gene cassettes [64,67], and the inappropriate antibiotic usage has produced a selective pressure and the consequential survival of resistant microorganisms (engendering multiple resistances).

Every year, bacterial genome sequencing allows the identification of emerging and re-emerging ARGs, and the most frequent examples are amplified from aquaculture seafood products, i.e., *sul* (sulfonamides resistance genes), *tet* (tetracyclines resistance genes), *aa* (aminoglycosides resistance genes), and *bla* (β-lactams resistance genes) [68,69,70,71]. Indeed, molecular biology, through the sequencing assays, constantly discovers different mutations among ARGs. There are numerous cases in which there is no matching between discovered phenotypic resistances results with the genotypic ones. This sentence offers explanations based on the concept of nucleotide sequences’ mutations that produce different DNA transcriptions (improper enzymes’ actions), or it is possibly correlated to an intrinsic resistance, which is typical of certain bacterial families against specific antibiotic molecules or classes. The scientific community, during these years, has investigated the AMR diffusion in various finfish species, especially in aquaculture systems, and the respective numerous genera of pathogenic and opportunistic bacteria that are generally implicated in seafood-borne diseases are *Vibrio* spp. (i.e., *V. parahaemolyticus, V, vulnificus*), *Listeria monocytogenes*, *Clostridium botulinum*, *Aeromonas* spp., *Salmonella* spp., *Escherichia coli*, *Campylobacter jejuni*, *Shigella* spp., *Yersinia eneterocolitica*, *Bacillus cereus* [41], *Pseudomonas* spp. [72], and *Enterococcus faecium* [73] (see Table 4). As previously mentioned, quinolones, tetracyclines, amphenicol, and sulfonamides are major antimicrobial classes used in aquaculture on a global scale [72].

### 5.1. Quinolones

Quinolone resistances are characterized by the involvement of *DNA gyrase* and *topoisomerase IV*, which are bacterial enzymes and quinolones target proteins. These two enzymes are encoded, respectively, by the *gyr*A and *gyr*B genes for *DNA gyrase*, and by the *par*C and *par*E genes for topoisomerase IV [38]. Chromosomal mutations in topoisomerases genes decrease drug accumulation and possible resistance driven by mobile elements, such as plasmid-mediated quinolone resistance (PMQR) (*Qnr* proteins, *aac(6)-lb-cr* aminoglycoside acetyltransferases and *QepA* and *OqxAB* efflux pumps), causing the constitutive or the acquired resistance to these antibiotic molecules. Increased mutations in *DNA gyrase* and *topoisomerase IV*, and in quinolone-resistant fish pathogens (*Yersinia ruckeri*, *Flavobacterium psychrophilum*, and *V. anguillarum*), are linked to the extensive administering of these antimicrobic classes worldwide [86,87,88]. Their wide usage was justified to reduce the hatching losses caused by *Vibrio* spp. infectious outbreaks. The wide detections of modified plasmids have been discovered from aquaculture finfish fillets [89]. The detected bacterial pathogens were *A. hydrophila*, *V. anguillarum*, and *V. parahaemolyticus,* which showed mutations in the quinolone resistance codifying sequences in specific gene regions belonging to *gyr*A and/or *par*C [90,91].

Quinolone resistance genes included in the so-called PMQR are: six *qnr* genes (*qnr*A, *qnr*B, *qnr*C, *qnr*D, *qnr*S, and *qnr*VC) encoding gyrase-protection repetitive peptides; *oqx*AB, *qep*A, and *qaq*BIII encoding efflux pumps; and *aac(60)-Ib-cr* encoding an aminoglycoside and quinolone inactivating acetyl-transferase [92]. The majority of PMQRs detection was largely amplified from finfish products worldwide; in China Yan et al. [93] found *qep*A and *aac-(6′)-Ib* genes as dominant among *PMQR genes* in aquatic environments and the possibility of co-emergence of resistance to β-lactams; Jiang and coworkers [94] detected *qnr*B, *qnr*S, and *qnr*D, with *aac(6′)-Ib-cr* in gut samples of farmed fish. Dobiasova et al. [95] found *qnr*S2, *aac(6)-Ib-cr* or *qnrB17* genes in *Aeromonas* spp. isolated from tropical freshwater ornamental fish and coldwater ornamental (koi) carps. In Egypt, scientists reported the occurrence of *qnr* and *aac(6)-Ib-cr* resistance from fish farm water sample [1].

In Chile, genes *qnr*A, *qnr*B, and *qnr*S were detected both in the chromosomes of marine bacteria and the same genes in human pathogenic ones [96]. Furthermore, it has been demonstrated that the same plasmid plays an important role for different classes. Indeed, gene cassettes can be considered as multiple ARGs drivers, which conduce to the phenotypical expression of resistance against quinolones, β-lactams, and aminoglycosides [92,93]. Indeed, *qnr* genes are loaded with β-lactamase determinants on the same plasmids. Khajanchi and coworkers [97] considered aquaculture and the aquatic environment as possible sources of *aac(6)-Ib-cr* and *qnrB2* and *Enterobacteriaceae* as hosts. They also detected Aeromonas spp. as a vector for *qnrS2*. Hence, Gram-negative hosts may a be reservoir of plasmid-mediated *Qnr-like* determinants that seem closely relate to the species *V. splendidus* [98].

From an environmental perspective, there is a strict correlation between remarkable anthropic activities as polluted water areas and quinolones ARGs diffusion [99]. Indeed, in Asia (especially in China), fish farmers normally use biofertilizers to improve production [100]. There is a real possibility that these organic molecules are vectors of antibiotic resistance genes. Zhao et al. [101] examined biofertilizers normally used in Chinese shrimp aquaculture systems and studied the correlation between fluoroquinolone resistance genes’ diffusion and biofertilizers. In this research project, they also screened the *PMQR gene* that includes: *qnr*A, *qnr*B, *qnr*C, *qnr*D, *qnr*S, *qep*A, *oqx*A, *oqx*B, and *aaa(6′)-Ib* genes. They screened 20 biofertilizer samples collected from shrimp farms and isolated 20 bacterial strains that were vectors of *PMQR genes*: 10 *Escherichia coli*, 9 *Enterococcus faecalis*, and 1 *Enterococcus faecium*. About 30% of biofertilizers samples presented *qnr*B, *qnr*D, and *qep*A resistance genes. This study was the first one which discovered the ARGs environmental repercussions due to the usage of contaminated manures on seafood farming systems. Similar patterns were also observed in terrestrial mammals, i.e., domestic swine and chicken manure (widely used in agriculture) [101]. Nowadays, there are not available data regarding finfishes, and it could be interesting to perform further investigations about possible statistical correlations between farming environments and possible agricultural implications. Therefore, these studies have confirmed ARGs diffusion and circulation in different environments through the fecal bacteria detected in common biofertilizer molecules. From these data, it can be seen that quinolones have presented reasonable risks due to the increase of their therapeutic failure. Their inclusion in the Critically Important Antimicrobials list has attracted more attention from pharma surveillance organizations.

### 5.2. Tetracyclines

Tetracyclines action consists of reversibly binding the 70S ribosome of cells blocking protein synthesis [1]. They are largely used in human and animal treatment as broad-spectrum antimicrobials. For the first time in Japan, it was observed that their improper administering conducted to the discovery of high nucleotide similarities of tetracycline genes between isolated bacteria from finfish aquaculture and from human clinical facilities. The phylogenetic analysis confirmed the same origin [3].

Evolution has selected different strategical and survival pathways and, in particular, four strategies: efflux pumps activation, ribosomal protection inducing a limit to the access, ribosomal RNA mutations avoiding tetracycline molecules binding, and tetracycline inactivation through enzymes [102,103]. In finfish aquaculture products, *tet* group responsible for proton-dependent efflux pumps encoding was mainly associated with tetracycline resistance [104]. *Tet* genes have been detected in several bacterial strains isolated from different animal species located in various geographical regions. There are multiple examples: *tet*B, *tet*M, *tet*W were firstly isolated in the intestine and rearing water of red seabream (*Pagrus major*) [105]; *tet*A, *tet*B, *tet*E, *tet*H, *tet*l, *tet*34, *tet*35 and 10 others had unknown *tet* genes isolated from Chilean salmon (*Salmo salar*) farms [106]; furthermore, Higuera-Llanten and coworkers [107] also detected the presence of *tet*34, *tet*35, *tet*A, *tet*B, *tet*E, *tet*H, *tet*L, and *tet*M genes in the same matrixes. Among seafoods (including finfish and crustaceans), Concha et al. [108] discovered *tet*X gene in *Epilithonimonas* strains from rainbow trout (*Oncorhynchus mykiss*) and Han et al. [109] amplified, in shrimp samples, that the *tet*B gene was carried in a single copy plasmid, named *pTetB-VA1,* comprising 5162-bp. The whole genome analysis revealed that this plasmid consists of 9 ORFs (overlapping open reading frames) encoding tetracycline-resistant repressor proteins, transcriptional regulatory proteins, and transposases and showed a 99% sequence identity to other *tet* gene plasmids (*pIS04-68* and *pAQU2*). Furthermore, in terms of *tet* genes, with special regard to *tet*E, Agersø et al. [110] discovered *tet*E horizontal transmission between *Aeromonas* spp. and *Escherichia coli* strains, isolated from aquaculture Danish farms. *Tet*A gene diffusion has been demonstrated to be realized through plasmids and transposons named *Tn1721* and those that are *Tn1721-like*. Another similar example is represented by *Tn5706,* which is involved in *tet*H dissemination (amplified from *Moraxella* spp. and *Acinetobacter* spp. strains isolated from salmon farms) [111]. Due to the expanding of the AMR phenomenon among different bacterial strains, *tet* genes have been widely amplified from *Enterobacteriaceae* [112,113], *Photobacterium* spp., *Vibrio* spp., *Alteromonas* spp., *Pseudomonas* spp., and other marine commensal bacteria. Consequently, the possibility of transferring ARGs from marine microbiota to the human one is considered reasonable. Indeed, many biomolecular investigations have highlighted the possible cross-species ARGs transmission through the foodstuffs ingestion [102,114].

The wide oligonucleotide diversities, as described above, are expressions of the mass administering of tetracycline. Any mammalian zootechnic sectors (i.e., domestic swine) have improperly used this antibiotic class, inducing multiplication and genetic transmissions to the next generations of bacterial isolates (from pathogen to commensal strains, and vice versa).

### 5.3. Sulfonamides

In aquaculture, sulfonamides are commonly co-administered with trimethoprim, ormethoprim, and florfenicol [115]. The dihydropteroate synthase (DHPS) enzyme, in the folic acid pathway, represents the biochemical target reaction [114]. Sulfonamide’s resistance mechanisms derive from mutations in the chromosomal *folP* gene that provides varying degrees of trade-off between resistance and efficient folate synthesis, decreasing DHPS affinity for the antimicrobial molecule [114].

Among the discovered ARGs, four different *sul* gene determinants have been described to encode antibiotic resistance. *Sul*1 gene has been founded in class 1 integrons and linked to other resistance genes [116]; *sul*2 is associated with non-conjugative plasmids of the *IncQ* group and to large transmissible plasmids, such as *pBP1* [117]. *Sul*3 is characterized in the Escherichia coli conjugative plasmid *pVP440*; *sul*4 gene has been recently mobilized and phylogenetic inference pinpoints its putative origin as part of the folate synthesis cluster in the *Chloroflexi phylum* [118]. All described ARGs have a common action, which is represented by the reduction in strategical bacterial structural expression. The transmembrane architectures are widely involved in the cyto-chemical interaction between strains and antibiotic molecules.

The genome and proteome analyses revealed that a gene cluster, containing a flavin-dependent monooxygenase and a flavin reductase, is highly upregulated in response to sulfonamides action, as reported by Kim et al. [119]. Indeed, the biochemical analysis showed that the two-components (belonging to the monooxygenase system) were key enzymes for the initial sulfonamides cleavage. It was observed that the co-expression of the two-component system in *Escherichia coli* conferred decreased susceptibility to sulfamethoxazole, indicating that the genes encoding drug inactivating enzymes are potential resistance determinants. Comparative genomic analysis revealed that this cluster gene, containing sulfonamide monooxygenase (renamed as *sul*X) and flavin reductase (*sul*R), is highly conserved in genomic islands. These ones are shared among sulfonamide-degrading Actinobacteria, all of which also contained *sul*1-carrying class 1 integrons [119].

Sulfonamide’s ARGs distribution has been widely found in numerous fish and environmental specimens, i.e., Muziasari and coworkers [120] discovered *sul*1, *sul*2, and *intI*1 genes detection in all analyzed samples and the *dfr*A1 gene in most samples in aquatic farm sediment in the Baltic Sea [39]. Domínguez et al. [121] detected *sul*1, *sul*2, class 1 integron-integrase gene *intI*1, *dfr*A1, *dfr*A12, and *dfr*A14 from a salmon farm in Chile and revealed the occurrence of transferable integrons and *sul* and *dfr* genes among sulfonamide- and/or trimethoprim-resistant bacteria, as amplified from *Actinobacter* spp., *Bacillus* spp., *Proteus* spp., and *Pseudomanas* spp. isolates [88].

ARGs for sulfonamides resistance were also discovered in many commensal bacterial strains in Japanese mariculture areas [122], in Vietnamese freshwater farms [108], China Hainan, Guangdong, Tianjin, Hangzhou, Yantai, and Taihu Lake [113,123,124,125].

These last considerations highlight that environmental *stimuli* can be responsible for increased or reduced ARGs transcriptions. The deduction leads to the consideration that in the AMR phenomenon, “*the environment*” plays a crucial role, while human and animal health are only “*direct consequences*”.

### 5.4. Thiamphenicol and Florfenicol

Thiamphenicol and florfenicol belong to the amphenicol antibiotic class and have been largely administered in aquaculture farms. Due to the possible chemical residual persistence in finfish muscular tissues, various studies have demonstrated possible sanitary implications on humans, animals, and environments [48,126].

Focusing on risk-based approach (in accordance with the EU Reg. No. 852/2004 and No. 37/2010), the European agencies EFSA and EMA published maximum residue limits and respective daily intakes for final human consumers [48].

Veterinary practitioners normally treat infectious disease (caused by, i.e., *Vibrio* spp., etc.) and relative possible septicemia cases using the above-mentioned molecules [127]. These have pharma-dynamic synergic effects (binding the 50S ribosomal subunit) if they were coupled with other antibiotic classes as tetracyclines. Both molecules have become widely prescribed because they have broad spectrum effects and low costs [128].

Amphenicol illegal administering has induced an intense evolutive pressure, determining the spreading of resistant strains harboring florfenicol-resistance genes (FRGs). These FRGs are plasmid determinants and have presented high genetic trades (through horizontal transmission) across different bacterial phyla, identifying strong correlations (*p* values < 0.05), as observed by Zeng et al. [127].

Among amplified FRGs, *cat*, *cfr, cml*, *fex*A, *fex*B, *flor*B, and *optr*A have been discovered from animal origin food matrices (including finfish ones). Their biochemical actions are involved in several pathways, i.e., protein synthesis inhibition, exporter ability, methyltransferase activation, efflux pumps, etc. [129,130].

From a microbiological perspective applied to the veterinary clinical aspects, amphenicol administering has demonstrated biochemical repercussions on intestinal microbiota. It induces shifts among bacterial biodiversity acting as strong stressor [127].

This last consideration finds explanations from cyto-chemical interactions directly associated with the consequential expression of transmissible oligonucleotide sequences. Among the above-mentioned amphenicol-resistant determinants, the metagenomic technology, coupled with next generation sequencing, has identified multiple mutations on open reading frames regions, which encode resistant mechanisms, i.e., efflux pumps, new binding epitopes, etc. [126].

Innovative biomolecular technologies, combining thiamphenicol and florfenicol administering, has permitted us to reduce their respective dosages but preserve their therapeutic efficacy [131].

The notable ARGs heterogeneity and their extreme variabilities pose the basis for further diagnostic and *One Health* clinical challenges. Aminophenols, as with other previously mentioned antibiotic classes, are widely used in the aquaculture zootechnic sector. Therefore, it is mandatory to preserve their therapeutic actions.

## 6. Antibiotic Substitutions

### 6.1. Vaccination

FAO reports numerous administered vaccines against different bacterial or viral diseases among finfish species. The most frequently used provide seroconversion against *Vibrio salmonicida*, *Vibrio anguillarum*, *Photobacterium damselae*, *Aeromonas salmonicida*, *Yersinia ruckeri*, etc. [3].

Conversely, there are few vaccines for viral diseases, in which usage is highly recommended in marine finfish farms [132]. In the aquaculture farms, vaccines can be administered through different methods: injection, in bath, or through the orofecal route [3]. Injection, through the intraperitoneal route, provides powerful and durable protection, but, on the other hand, this procedure influences animal welfare, inducing a relevant stress condition. It is commonly used for *Salmo salar* finfish species but is not applicable for other species, i.e., *Pangasius* spp. and *Tilapia* spp. Conversely, oral administering reduces stress (due to animal handling), since animals receive immunization through food ingestion. The main difference between these two above-mentioned methods is represented by the need for large amounts of antigens in the ingestion method to obtain an adequate immunity [133]. There are contrasting opinions on vaccines’ efficacy regarding finfish farms. Usually, after vaccination, fish farmers must administer antibiotics to control infectious disease outbreaks [134]. This condition is related to an incomplete understanding of the vaccination type and the immune system’s reaction to the “*antigenic stimuli*”. It is improper to compare fish immune reactions with the generated response in mammalians [135].

However, in any species, such as farmed Atlantic salmons, vaccination represents an important preventive tool [3]. Farmed salmonids (*Salmo salar*) receive immune protection through the injection of a pentavalent vaccine against vibriosis, furunculosis, piscrickettsiosis, infectious pancreatic necrosis, and infection salmon anemia. The vaccine has permitted a reduced usage of antibiotics [135]. In tilapia’s farms, the mucosal administering route replaces the injective method. In this fish species, evidence supports a competent immune stimulation of the antigen-presenting cells (similarly to mammalians). In this way, fish farmers reduce antibiotic administering [136].

A new frontier is represented by nano-material vaccines, which use virus-like particles, immune-stimulating complexes, liposomes, polymeric, etc. These molecules drive antigens and can drive protective responses in fish. Furthermore, nanoparticles permit antigens’ release, and, for this reason, booster vaccinations are not necessary [137], but live attenuated vaccines’ employment in aquaculture is not allowed by the European Commission. Their usage has not yet been allowed due to the wide gap of knowledge concerning possible implications on human consumers [138].

### 6.2. Structural Improvements

Innovative production systems have become popular among fish farmers, i.e., catfish aquaculture in the USA [65]. New fish farming systems that provide more space and efficient wastewater management allow an avoidance of the large usage of antibiotic molecules [6].

Therefore, in the USA, fish farmers introduced an innovative system called “*spilt-pond*” to optimize fish’s sanitary conditions and productive levels. This new system is realized through the division of the traditional ponds in two areas: an algal growth basin and a fish holding area. In this way, the growth of production is allowed by the high animals’ density in the same period of production, and the reduction in antimicrobial use is due to the continuous water filtration [139]. Conversely, in Malaysia and other Asian-Pacific regions, fish are farmed by using pond culture, ex-mining pools, cement tanks, and freshwater pen culture systems. In these structures, there is low water filtration. Animal catabolites and feces remain for all productive cycles, producing a functional substrate for any bacterial species (i.e., Enterobacteriaceae) proliferation. Furthermore, in such countries of this continent (China, Vietnam, Philippines, India, etc.), the usage of antibiotics in aquaculture is not well regulated by national law [6]. Therefore, a new approach to the aquaculture systems of production is required. Indeed, Brunton et al. [140] generated a mapping system obtained through the stakeholders’ collaboration. Correlating ecological aspects to the new above-mentioned fish farms realities. It identifies hotspots and risk points related to antibiotic usage in the aquaculture food chain. The platform provides a quantitative risk analysis at different steps of production. Therefore, these maps allow us to understand the molecules’ flows, ARGs, and ARB. In this way, it is possible to monitor antibiotic resistance factors. From these elaborated data, food safety authorities may program control activities through surveillance measures.

### 6.3. Probiotics

Probiotics have different effects on fish farming issues, i.e., they reduce animal mortality (especially at the larval stage) [141,142,143], improve animal welfare through the immune system’s stimulation, and reduce the antibiotic therapies’ necessity. Fish farmers introduce these bacteria through the finfish diet, as supplementary feed [3]. *Bacillus* spp. Is largely used in numerous fish farms realities for its probiotic properties. This genus can mitigate pathogenic microorganisms’ growth and can eliminate ARB [143]. These capacities are related to the bioactive peptides’ synthesis (bacteriocin) [144], but there are also nonpeptidic molecules, i.e., phospholipids, polyketides, etc., that are classified as bacteriocin [145]. *Bacillus* spp. produces CAMT2; this molecule is a recent example of bacteriocin that inhibits the proliferation of different bacterial strains, i.e., *Vibrio* spp., *Staphylococcus aureus*, and *Listeria monocytogenes* [146]. Another interesting microbiological aspect of this genus is represented by bacterial competition. It includes competition for energy (obtained from substrates), nutrients, and adhesion sites [5]. Indeed, *Bacillus* spp. Can rapidly colonize organic substrates with strong adhesion capacities (due to hydrophobic and steric forces) [147]. Therefore, pathogenic bacteria find an inadequate micro-environment that results in hostility to their proliferation. Furthermore, *Bacillus* spp. also stimulates fish’s cell-mediated immune response. Indeed, bacterial pathogens decrease their virulence because the animal host presents a resistant and competent immune system [148].

Healthy animals require few antibiotic therapies, leading to the reduction in antibiotic consumption in the next ten years [3]. Thanks to these preventive measures, different finfish species (i.e., parrotfish) have become resistant to *Vibrio alginolyticus* infection. These considerations are strictly related to the concept that powerful immune systems reduce pathogen bacterial proliferation [60].

## 7. Conclusions

The AMR is a real public health issue [3]. This review aims to provide a current scenario about the circulation of ARGs in bacterial isolates, usually identified from aquaculture food industry chains. The authors also want to highlight that the AMR phenomenon is a dynamic concept, differing in time and regional areas among years. In this case, a particular and not limited ecosystem represents the main challenge. Oceans’, lakes’, rivers’ health requires periodical screenings.

Molecular biology has a key role in studying bacterial genomic mutations and ARGs horizontal transmission flows. In the future, marine currents’ studies will also have a crucial role filling any gaps of knowledge. They will allow the estimation of any geographical areas where ARGs will be widely diffused and improve the introduction of efficient corrective measures. In this way, a new concept of “*Environmental Medicine*” will produce factual results based on a holistic point of view.

Vaccination programs, probiotics’ administering, and structural improvements are three examples that represent valid alternative measures to reduce antibiotic usage.

These measures will produce satisfactory results if all nations of the world adopt them according to their ecosystems’ peculiarities.

These efforts are necessary due to the growing demand for animal origin finfish proteins and the increasing global demographic. However, these physiological necessities must be satisfied involving sustainable and innovative production systems.

## Figures and Tables

**Table 1 antibiotics-11-01574-t001:** Pharmacologically active substances and their classification regarding maximum residue limits (MRL) in foodstuffs of animal origin (from EU Reg. No. 37/2010).

Pharmacologically Active Substance	Marker Residue	Animal Species	MRL*
Benzylpenicillin	Benzylpenicillin	All other food-producing species.	50 µg/kg
Chlortetracycline	Sum of parent drug and its 4-epimer	Fin fish (all other food-producing species).	100 µg/kg
Cloxacillin	Cloxacillin	Fin fish (all other food-producing species).	300 µg/kg
Colistin	Colistin	Fin fish (all other food-producing species).	150 µg/kg
Danofloxacin	Danofloxacin	Fin fish (all other food-producing species).	100 µg/kg
Dicloxacillin	Dicloxacillin	Fin fish (all other food-producing species).	300 µg/kg
Difloxacin	Difloxacin	Fin fish (all other food-producing species).	300 µg/kg
Enrofloxacin	Enrofloxacin	Fin fish	100 µg/kg
Erythromycin	Erythromycin A	Fin fish	200 µg/kg
Florfenicol	Sum of florfenicol and its metabolites measured as florfenicol amine	Fin fish	1000 µg/kg
Flumequine	Flumequine	Fin fish	600 µg/kg
Lincomycin	Lincomycin	Fin fish (all other food-producing species).	1000 µg/kg
Neomycin (including Framycetin)	Neomycin B	Fin fish (all other food-producing species).	500 µg/kg
Oxacillin	Oxacillin	Fin fish (all other food-producing species).	300 µg/kg
Oxolinic acid	Oxolinic acid	Fin fish (all other food-producing species).	100 µg/kg
Oxytetracycline	Sum of parent drug and its 4-epimer	Fin fish (all other food-producing species).	100 µg/kg
Paromomycin	Paromomycin	Fin fish (all other food-producing species).	500 µg/kg
Sarafloxacin	Sarafloxacin	Salmonidae	30 µg/kg
Spectinomycin	Spectinomycin	Fin fish (all other food-producing species).	300 µg/kg
Sulfonamides (all substances belonging to the Sulfonamides group)	Parent group	Fin fish (all other food-producing species).	100 µg/kg
Tetracycline	Sum of parent drug and its 4-epimer	Fin fish (all other food-producing species).	100 µg/kg
Thiamphenicol	Thiamphenicol	Fin fish (all other food-producing species).	50 µg/kg
Tilmicosin	Tilmicosin	Fin fish (all other food-producing species).	50 µg/kg
Trimethoprim	Trimethoprim	Fin fish (all other food-producing species).	50 µg/kg
Tylosin	Tylosin	Fin fish (all other food-producing species).	100 µg/kg

***MRL**: Maximum residue limit. It represents the length of time necessary to assure the absence or below-defined values of drug molecules in animals’ tissues. Target tissue: muscle (related to “muscle and skin”, as reported by art.14(7) EU Reg. No. 470/2009). Therapeutic classification: Anti-infectious agents/antibiotics.

**Table 2 antibiotics-11-01574-t002:** FDA-approved aquaculture drugs.

Antimicrobials/Chemical Molecules	Use	Dose	Withdrawal Time and Other Limitations (Useful for MRL*)
Chloramine-T	For the control of mortality in: Freshwater-reared salmonids infected by *Flavobacterium* spp. Walleye due to *Flavobacterium columnare*.	12–20 mg/L (administered as a static bath every day for three treatments).	0 day
Formalin (37%)	The use of formalin is possible to be expanded as a parasiticide for all finfish and penaeid shrimp and as a fungicide to the eggs of all finfish	Administered in tanks and raceways for up 1 h (µL/L): Salmon and trout → up to 170 µL/L with a temperature above 10 °C/50 °F, or → up to 250 µL/L with a temperature below 10 °C/50 °F. All other finfish → up to 250 µL/L.	0 day
Hydrogen peroxide (35%)	For the control of mortality in finfish’s eggs and other losses caused by *Flavobacterium branchiophilum* and *F. columnare*.	Freshwater-reared finfish eggs: 500 to 1000 mg/L for 15 min in a continuous flow system (consecutive or alternate days) until hatch. Freshwater-reared salmonids: 100 mg/L for 30 min or 50–100 mg/L for 60 min once per day on alternate days for three treatments in a continuous flow.	0 day
Oxytetraycline hydrochloride	For the marking of skeletal tissues in finfish fry and fingerlings.	200–700 mg/2 L of water for 2 to 6 h.	0 day
Florfenicol	For control mortality caused by *Edwardsiela ictaluri* (enteric septicemia) and *Flavobacterium columnare*.	10 mg/kg of body weight for 10 consecutive days.	12 days (under veterinarian prescription)
Oxytetracycline dehydrate	Control *Aeromonas liquifaciens* and *Pseusomonas* spp. disease (they cause hemorrhagic septicemia), especially in *Oncorhynchus* spp. and *Salmo* spp.	10 mg/kg of body weight for 10 consecutive days.	21 days to catfish and 30 days to lobster
Sulfadimethoxine/ormetoprim	Control of *E. ictulari*	50 mg/kg of body weight for 5 days.	3 days.

***MRL**: Maximum residue limit. It represents the length of time necessary to assure the absence or below-defined values of drug molecules in animals’ tissues.

**Table 4 antibiotics-11-01574-t004:** AMR* and MDR* of bacterial strains isolated from aquaculture finfish sample tissues.

Country	Finfish Samples n.	Isolated Bacterial Strains	Phenotypic AMR*/MDR*	References
Brazil	n. 101 *Oreochromis niloticus*	*Salmonella* spp. (46 isolates)	Amoxicillin/Clavulanic acid (87.7%) Tetracycline (82.5%) Sulfonamide (57.9%) Chloramphenicol (26.3%) 56:1% of *Salmonella* spp. isolates were MDR: Beta-lactam (*bla*_CTX_ gene 66.7%) Tetracycline (*tet*A gene 54.4%) Chloramphenicol (*floR* gene 50.9%) Sulfonamide (*sul*2 gene 49.1%)	[74]
	n. 50 *Cyprinus carpio * n. 50 *Oreochromis niloticus*	*Enterococcus faecalis* (79 isolates)	Tetracycline (57.7% *tet*L and *tet*M) Erythromycin (31.01% *msr*C)	[75]
China	n. 50 fish samples: *Aristichthys nobilis* *Carassius auratus* *Ctenopharyngodon idellus* *Parabramis pekinensis*	*Vibrio cholerae* (370 isolates)	MDR: Streptomycin (62.2%) 230 Ampicillin (60.3%) 223 Rifampicin (53.8%) 199	[76]
	n. 17 *Acipenser* spp.	*Streptococcus iniae* (18 isolates)	Tetracycline (35.6% *tet*A-02) Beta-lactams (25.3% *bla*_TEM_) Aminoglycosides (22.1% *aad*A1)	[77]
	n. 75 *Carassius auratus*	*Aeromonas hydrophila* (n. 28 isolates)	MDR: Penicillin (100%) Ampicillin (100%) Amoxicillin (96.4%) Piperacillin (92.9%) Cefalexin (78.6%) Doxitard (75%) Teicoplanin (67.9%)	[78]
India	n. 25 *Oreochromis niloticus*	*Pseudomonas entomophila* *Aeromonas hydrophila*	MDR: Bacitracin (100%) Ampicillin (70%) Cephalothin (60%) Cafazolin (50%) All resistant to: Amoxicillin Ampicillin	[79]
	n. 97 *Mugil cephalus*	*Listeria monocytogenes* (n. 21 isolates)	69% of *Listeria* isolates were MDR to: Ampicillin Penicillin Erythromycin Tetracycline Clindamycin	[80]
Armenia	n. 25 *Oncorhyncus mykiss*	*Pseudomonas* spp.: *P. anguilliseptica* *P. fluorescens* *P. stutzeri* *P.putida* *P. aeruginosa* *P. algaligenes*	Resistance percentages: Piperacillin (45.6%) Pefloxacin (33.3%) Ciprofloxacin (3.2%) All susceptible to: Chloramphenicol	[72]
Italy	n. 300 fish samples: n. 100 *Dicentrarchus labrax* n. 100 *Umbrina cirrose* n. 100 *Sparus aurata*	*Vibrio* spp. *Aeromonas* spp. *Shewanella* spp. *Photobacterium* spp.	Resistance percentages: Tetracycline (11.54%) (147/1274) Trimethoprim/Sulfadiazine (7%) (89/1274)	[81]
Vietnam	n. 50 *Ictalurus* spp.	*Pseudomonas* spp. (n. 116 isolates)	Ampicillin (99.1%) Sulfamethoxazole (93.1%) Chloramphenicol (88.8%) Nitrofurantoin (90.5%) Nalidixic acid (90.5%) Norfloxacin (9.5%) Ciprofloxacin (8.6%) Tetracycline (30.2%) Doxycycline (25%)	[82]
		*Aeromonas* spp. (n. 92 isolates)	Ampicillin (93.5%) Sulfamethoxazole (60.9% Chloramphenicol (31.5%) Nitrofurantoin (25%) Nalidixic acid (52.2%) Ciprofloxacin (7.6%) Norfloxacin (4.4%)	
Vietnam Scotland Denmark Norway France Bangladesh Thailand Indonesia Ecuador	n. 44 fish samples: n. 12 *Pangasiodon hypophthalmus* 11 *Salmo salar* 10 *Crassostrea gigas* 11 *Penaeus mongodon*	*Escherichia coli* (n. 60) *Enterococcus* spp. (n.69) *Pseudomonas* spp. (n. 26) *Staphylococcus aureus* (n. 9) (246 isolates)	MDR strains: n. 7 *E. coli* resistant to: Chloramphenicol Ciprofloxacin Ampicillin Nalidixic acid Sulfamethoxazole Trimethoprim n. 3 *Enterococcus faecalis* resistant to: Chloramphenicol Gentamicine Tetracycline n. 4 *Staphylococcus aureus* resistant to: Chloramphenicol Kanamycin Tetracycline	[83]
Côte d’Ivoire	n. 480 *Oreochromis niloticus*	n. 1696 strains: *Escherichia coli* (15.9%) *Pseudomonas aeruginosa* (10.4%) *Bacillus cereus* (14.9%) *Enterococcus faecalis* (14.2%) *Citrobacter freundii* (13.5%)	Resistance percentages: Amoxicillin/Clavulanic Acid (5.8%) Piperacillin and Penicillin (8.7%) Gentamycin (7.2%)	[84]
Iran	n. 240 *Trota iridea*	n. 86 *Listeria* spp. isolates	Tetracycline (62.79%) Enrofloxacin (56.97%) Ciprofloxacin (38.37%) Penicillin (36.04%) Ampicillin (34.88%)	[85]

*AMR: Antimicrobial resistance. *MDR: Multidrug resistance.

## Data Availability

Not applicable.

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
