# Peer review of "Antibiotic Resistance in the Finfish Aquaculture Industry: A Review"

_antibiotics, 2022, doi:10.3390/antibiotics11111574_

Round 1

Reviewer 1 Report

I like the scope and organization of this review. However, effective use of language and excellent grammar are especially important for producing clear and comprehensive review articles. I would suggest retaining the structure of the paper and reexamining the writing throughout.

Author Response

Dear Reviewer 1,

All Authors want to express their appreciation to consider our manuscript for publication. Reviewer suggestions resulted precious and important to improve its quality and their observation were useful to implement the Scientific impact of our paper.

In the following two paragraphs we reported all changes:

Reviewer 1 suggestions: I like the scope and organization of this review. However, effective use of language and excellent grammar are especially important for producing clear and comprehensive review articles. I would suggest retaining the structure of the paper and reexamining the writing throughout.

Authors: Following the above-mentioned suggestions, a deep grammar language revision was performed.

We really appreciate all provided observations and hope that our manuscript has increased its scientific values.

Thank You for Your time and attention.

Best regards,

Gianluigi Ferri

Doctor in Veterinary Medicine (D.V.M.)

Ph.D. Student in Food Inspection

Faculty of Veterinary Medicine; University of Teramo, Italy.

Reviewer 2 Report

Dear authors,

The manuscript provides an overview of antimicrobial resistance in finfish aquaculture industry. The following points are addressed by the authors: legislative measures, sources of antimicrobial resistance and dissemination into aquatic environments, global usage of antibiotics in the aquaculture industry and the most prevalent antibiotic classes administrated in the finfish farms, antibiotic resistant bacteria and antibiotic resistance genes detected in the aquaculture sector. The material is well documented and structured, however some sections need be revised.

Please find below my suggestions and comments:

There are paragraphs that contain only one sentence. A paragraph is formed from at least 2-3 sentences.

Lines 29-30: add the number, i.e 156 million tonnes

Line 71: add WHO before CIA

Line 74: replace antimicrobic with antibiotic

Line 94: define AMR

Lines 225-228: define high amounts (higher than what); high amounts of tetracycline or high amounts of genes coding for tetracycline resistance? Same questions for the other antibiotics mentioned in this section

Lines 228-231: this section is confusing for me; horizontal gene transfer mechanisms were described between bacterial species, not animals.

Lines 232-235: define antibiotic resistant forms? Please clarify the correlation between fish food matrices harbouring ARB and ARG and nosocomial microorganisms.

Lines 274-279: Please define the term MRA = multi-resistant bacteria

Lines 280-334: (Section 4). Global antibiotic production and administration in the aquaculture sector), please include a table or a graph illustrating the main classes of antibiotics used and the amount in different geographical region. The title of this section should be modified, as the content is describing the usage of antibiotics not production. Also, please provide a definition of the term MDR.

Line 336: AMR is referring to microorganisms (i.e fungi, protozoa, viruses) resistant to different substances. The authors might consider also changing the article title? If they are referring only to antibiotics and bacteria.

Lines 351-354: there are several genes tet, sul, bla, therefore use plural for each category of genes

Thank you very much

Kind regards

Author Response

Dear Reviewer 2,

All Authors want to express their appreciation to consider our manuscript for publication. Reviewer suggestions resulted precious and important to improve its quality and all observation were useful to implement the Scientific impact of our paper.

In the following paragraphs we reported all changes:

Reviewer 2 suggestions:

Dear authors,

The manuscript provides an overview of antimicrobial resistance in finfish aquaculture industry. The following points are addressed by the authors: legislative measures, sources of antimicrobial resistance and dissemination into aquatic environments, global usage of antibiotics in the aquaculture industry and the most prevalent antibiotic classes administrated in the finfish farms, antibiotic resistant bacteria and antibiotic resistance genes detected in the aquaculture sector. The material is well documented and structured; however some sections need be revised.

Please find below my suggestions and comments:

Reviewer 2: There are paragraphs that contain only one sentence. A paragraph is formed from at least 2-3 sentences.

Authors (Line: 1564): This paragraph has been removed.

Reviewer 2 (Lines 29-30): add the number, i.e 156 million tonnes

Authors (Lines 29-30): This sentence has been revised following reviewer's suggestion.

Reviewer 2 (Line 71): add WHO before CIA

Authors (Line 185): WHO has been added before CIA.

Reviewer 2 (Line 74): replace antimicrobic with antibiotic.

Authors (Line 188): antimicrobic has been replaced with antibiotic.

Reviewer 2 (Line 94): define AMR

Authors (Line 209): AMR has been defined.

Reviewer 2 (Lines 225-228): define high amounts (higher than what); high amounts of tetracycline or high amounts of genes coding for tetracycline resistance? Same questions for the other antibiotics mentioned in this section.

Authors (Lines 520-523): This sentence has been clarified.

Reviewer 2 (Lines 228-231): this section is confusing for me; horizontal gene transfer mechanisms were described between bacterial species, not animals.

Authors (Lines: 523-527): This section has been clarified.

Reviewer 2 (Lines 232-235): define antibiotic resistant forms? Please clarify the correlation between fish food matrices harbouring ARB and ARG and nosocomial microorganisms.

Authors (Lines 528-537): These concepts have been clarified.

Reviewer 2 (Lines 274-279): Please define the term MRA = multi-resistant bacteria.

Authors (Lines: 751-754): The term MRA has been replaced with MDR and the relative concepts have been better clarified.

Reviewer 2 (Lines 280-334): (Section 4). Global antibiotic production and administration in the aquaculture sector), please include a table or a graph illustrating the main classes of antibiotics used and the amount in different geographical region. The title of this section should be modified, as the content is describing the usage of antibiotics not production. Also, please provide a definition of the term MDR.

Authors suggestions:

(Line 745): MDR has been defined.

(Line 750): Section title has been modified.

(Line: 909): A Table (Table 3) has been introduced.

Reviewer 2 (Line 336): AMR is referring to microorganisms (i.e fungi, protozoa, viruses) resistant to different substances. The authors might consider also changing the article title? If they are referring only to antibiotics and bacteria.

Authors (Line 1): Article title has been changed.

Authors (Lines: 910-914): This sentence has been revised following reviewer’s suggestion.

Reviewer 2 (Lines 351-354): there are several genes tet, sul, bla, therefore use plural for each category of gene.

Authors (Lines: 1124-1125): this sentence has been modified.

We really appreciate all provided observations and hope that our manuscript has increased its scientific values.

Thank You for Your time and attention.

Best regards,

Gianluigi Ferri

Doctor in Veterinary Medicine (D.V.M.)

Ph.D. Student in Food Inspection

Faculty of Veterinary Medicine; University of Teramo, Italy.

Reviewer 3 Report

1. The topic of the manuscript was antimicrobial resistance in the finfish aquaculture, but it did not review around "finfish aquaculture " and " antimicrobial resistance". The manuscript mentioned the drug resistance of shrimp bacteria many times, and more related to antimicrobial usage regulations and preventive measures (it is more appropriate to called it antibiotic substitution), which was a little inconsistent with the topical subject.

2. Since September 1, 2015, norfloxacin has been banned in aquaculture in China.

3. In Table 3, it is inappropriate to quote only one article to reflect the antimicrobial resistance of aquaculture of a country or region.

4. In “5. Antibiotic resistant bacteria (ARBs) and genes (ARGs) in finfish aquaculture”, it should be the key part of the manuscript, only the ARGs of quinolones, tetracyclines and sulfonamides were reiviewed. Other types of antibacterial drugs were also mentioned in Table 1 and Table 2, such as Florfenicol (Table 1) and Thiamphenicol (table 2) , its ARGs should also be introduced.

5. The authors just listed the ARGs of quinolones, tetracyclines and sulfonamides, but did not summarize these contents and put forward their own views.

6. There were many English grammar problems.

Author Response

Dear Reviewer 3,

All Authors want to express their appreciation to consider our manuscript for publication. Reviewer’s suggestions resulted precious and important to improve its quality and their observation were useful to implement the Scientific impact of our paper.

In the following paragraphs we reported all changes:

Reviewer 3 suggestions:

Reviewer 3: The topic of the manuscript was antimicrobial resistance in the finfish aquaculture, but it did not review around "finfish aquaculture " and " antimicrobial resistance". The manuscript mentioned the drug resistance of shrimp bacteria many times, and more related to antimicrobial usage regulations and preventive measures (it is more appropriate to called it antibiotic substitution), which was a little inconsistent with the topical subject.

Authors (Lines: 1240-1243): The considerations about biofertilizers as possible ARGs’ drivers have been explained.

Authors (Line 1513): Paragraph’s title has been modified.

Authors (Lines 1204-1221): Manuscript has been implemented following reviewer suggestions.

Drug resistance about shrimps was cited because it was the first study which correlated biofertilizers usage to the antibiotic resistance determinants diffusion.

Reviewer 3: Since September 1, 2015, norfloxacin has been banned in aquaculture in China.

Authors (Line: 389): Following Reviewer’s suggestion “norfloxacin” has been removed.

Authors (Lines: 929-930): This concept has been introduced.

Reviewer 3: In Table 3, it is inappropriate to quote only one article to reflect the antimicrobial resistance of aquaculture of a country or region.

Authors (Table 4): Table 4 has been implemented following Reviewer’s suggestion.

Reviewer 3: In “5. Antibiotic resistant bacteria (ARBs) and genes (ARGs) in finfish aquaculture”, it should be the key part of the manuscript, only the ARGs of quinolones, tetracyclines and sulfonamides were reiviewed. Other types of antibacterial drugs were also mentioned in Table 1 and Table 2, such as Florfenicol (Table 1) and Thiamphenicol (table 2), its ARGs should also be introduced.

Authors (Lines: 1375-1412): A paragraph, regarding thiamphenicol and florfenicol, has been introduced.

Reviewer 3: The authors just listed the ARGs of quinolones, tetracyclines and sulfonamides, but did not summarize these contents and put forward their own views.

Authors (Lines: 1157-1170): This paragraph has been modified following reviewer’s suggestion.

Authors (Lines: 1211-1221): This paragraph has been modified following reviewer’s suggestion.

Authors (Lines: 1260-1267): This paragraph has been modified following reviewer’s suggestion.

Authors (Lines: 1298-1302): This paragraph has been modified following reviewer’s suggestion.

Authors (Lines: 1339-1342): This paragraph has been modified following reviewer’s suggestion.

Reviewer 3: There were many English grammar problems.

Authors: Following reviewer’s suggestion, a deep grammar language revision was performed.

We really appreciate all provided observations and hope that our manuscript has increased its scientific values.

Thank You for Your time and attention.

Best regards,

Gianluigi Ferri

Doctor in Veterinary Medicine (D.V.M.)

Ph.D. Student in Food Inspection

Faculty of Veterinary Medicine; University of Teramo, Italy.

Round 2

Reviewer 1 Report

The authors appear to have carefully addressed reviewers comments in this updated draft, and it is improved. However, the writing is still in need of reconsideration. I do apologize for the legibility of the attached draft--I had absent-mindedly begun reworking the abstract before I realized it was getting a bit messy. I did make limited and hopefully more readable notes throughout. I would highly recommend to the authors that they review their paragraph structure and perhaps go sentence by sentence to make sure their points are coming across as clearly and in as engaging a manner as possible.

Author Response

Dear Reviewer,

We are grateful for the provided suggestions. Following Your indications, we think that our manuscript has improved its scientific quality.

In the following paragraphs we reported all changes:

Reviewer suggestions: The authors appear to have carefully addressed reviewers comments in this updated draft, and it is improved. However, the writing is still in need of reconsideration. I do apologize for the legibility of the attached draft--I had absent-mindedly begun reworking the abstract before I realized it was getting a bit messy. I did make limited and hopefully more readable notes throughout. I would highly recommend to the authors that they review their paragraph structure and perhaps go sentence by sentence to make sure their points are coming across as clearly and in as engaging a manner as possible.

Authors (lines 9-21): Following reviewer’s suggestions (.pdf file), the Abstract section has been revised.

Authors (lines 25-44): Following reviewer’s suggestions (.pdf file), this part has been revised.

Authors (lines 234-245): Following reviewer’s suggestions (.pdf file), this part has been revised.

Authors (line 313): Following reviewer’s suggestions (.pdf file), this part has been revised.

Authors (line 316): Following reviewer’s suggestions (.pdf file), this part has been revised.

Authors (lines 329-332): This section is part of Table 1 used for the explnation of MDR acronym.

Authors (lines 331-344): Following reviewer’s suggestions (.pdf file), this part has been revised.

Authors (lines 364-371): Following reviewer’s suggestions (.pdf file), this part has been revised.

Authors (line 384): Following reviewer’s suggestions (.pdf file), this part has been revised.

Authors (line 472): Following reviewer’s suggestions (.pdf file), this part has been revised.

Authors (lines 521-523): Following reviewer’s suggestions (.pdf file), this part has been revised.

Authors (lines 565-566): Following reviewer’s suggestions (.pdf file), this part has been revised

Authors (lines 593-594): Following reviewer’s suggestions (.pdf file), this part has been revised

Authors (lines 761-795): Following reviewer’s suggestions (.pdf file), this part has been revised

Authors (lines 814-967): Following reviewer’s suggestions (.pdf file), this part has been revised

Authors (lines 1064-1090): Following reviewer’s suggestions (.pdf file), this part has been revised

Authors (lines 1093-1110): Following reviewer’s suggestions (.pdf file), this part has been revised

We really appreciate all provided observations and hope that our manuscript has increased its scientific values.

Thank You for Your time and attention.

Best regards,

Gianluigi Ferri

Doctor in Veterinary Medicine (D.V.M.)

Ph.D. Student in Food Inspection

Faculty of Veterinary Medicine; University of Teramo, Italy.
